# Antifungal susceptibility of *Candida* species to copper oxide nanoparticles on polycaprolactone fibers (PCL-CuONPs)

**Antonio Muñoz-Escobar, Simón Yobanny Reyes-López**[ORCID]*

Instituto de Ciencias Biomédicas, Universidad Autónoma de Ciudad Juárez, Envolvente del PRONAF y Estocolmo s/n, Ciudad Juárez, Chihuahua, México

* simon.reyes@uacj.mx

**Data Availability Statement:** All relevant data are within the paper.

**Funding:** The author(s) received no specific funding for this work.

## Abstract

The integration of metallic or ceramic nanoparticles in polymer matrices has improved the antimicrobial and antifungal behavior, resulting in the search for composites with increased bactericidal and antimycotic properties. A polycaprolactone fibers with copper oxide nanoparticles was prepared. Polycaprolactone-copper fibers (PCL- CuONPs) were prepared into two major steps in situ method: (a) Synthesis of CuO particles, then (b) incorporation of polycaprolactone to electrospun process. The first step is the reduction of Cu+2 ions by gallic acid in N,N-dimethylformamide and tetrahydrofuran solution with the simple addition of polycaprolactone in the solution for the second electrospun step. Raman spectra provide information about the nature of the copper oxide synthesized. There are three Raman peaks in the sample, at 294 and 581 cm$^{-1}$ and a very broad band from 400 to 600 cm$^{-1}$ which are characteristics bands for CuO. Scanning electron microscopy (TEM) revealed copper oxide nanoparticles with semispherical shapes with diameter 35 ±11 nm. Dynamic light scattering (DLS) analysis showed uniform CuONPs in a range of 88±11 nm. Scanning electron microscopy (SEM) of PCL-CuONps reveled fibers with diameters ranging from 925 to 1080 nm were successfully obtained by electrospinning technique. Orientation, morphology and diameter were influenced by the increment on CuONPs concentration, with the smaller diameter present in samples prepared from low concentrated solutions. The antimycotic applicability of the composite was evaluated to determine the antifungal activity in three species of the genus Candida (*Candida albicans*, *Candida glabrata* and *Candida tropicalis*). PCL-CuONPs exhibit a considerable antifungal effect on all species tested. The preparation of PCL-CuONPs was simple, fast and low-cost for practical application as an antifungal dressing.

## Introduction

The fast emergence of antibiotic and antifungal resistant bacteria and fungi is occurring on a world scale. The overuse and misuse of these medications, as well as a lack of new antibacterial and antimycotic materials have been the cause of this antibiotic and antifungal resistance

**Competing interests:** The authors have declared that no competing interests exist.

crisis. The most notable of the diseases present in the oral cavity in humans is a common fungal infection known as oral candidiasis. This is an opportunistic infection which may reflect immunological changes and a possible association with potentially malignant disorders of the oral mucosa. Previous studies have demonstrated that 80% of clinical isolates are produced by *Candida albicans*, and almost 15% of all Candida-related systemic bloodstream infections are related to *Candida glabrata* [1]. Several other Candida species, like *C. tropicalis*, *C. krusei* and *C. parapsilosis* have increased significantly the number of infections in the last decade, especially *C. tropicalis* which is has become the most common non-albicans Candida species in Asia [2,3].

Recently, the increment of cases of opportunistic infections of oral candidiasis among immunocompromised hosts resulting from transplant, hematologic malignancies, hemodialysis, cancer chemotherapy, or organ or human immunodeficiency virus infections [4], have prompted researchers to study antimicrobial agents due to they can vary in their susceptibility to common prescribed antifungal agents even within the same species [5]. The ability of genetic and phenotypic adaptation of *C. albicans* is well known, allowing it to improve its survival strategies especially in the oral cavity when the immune response is deficient and the response of tissues is altered as a primary barrier, reason for which it is considered of great importance the research in products that inhibit the growth of this microorganism [6].

The use of nanotechnology is a viable prospective that can be employed in order to address this issue effectively [7]. Nanomaterials based on metal ions, have an outstanding of cytotoxicity activity against bacteria, fungi and viruses. Nanometallic materials display this antimicrobial property due to their surface change and surface to volume ratio, which allows an enhanced dilution of metal ions disabling enzymes and DNA of microorganisms creating an electron imbalance between donor groups such as thiol, carboxyl, and hydroxyl groups [8].

Copper oxide nanoparticles (CuONPs) stand out by possessing very well-known antibacterial properties, being active against a varied range of pathogenic bacteria [9]. The administration of CuONPs as an antimicrobial treatment needs a vehicle, being the polymer films the most widely used materials in this area. The integration of metal oxide nanoparticles improves its antimicrobial as well as its physical, chemical and mechanical properties [10].

Electrospinning is an effective technique to produce polymer biofilms, which allows the fabrication of continuous filaments at a nano and microscale from natural and synthetic polymers, in this manner increasing the possibility to be like biological and mechanical properties [11]. Polycaprolactone (PCL) is one of the most broadly employed polymers in biomedical applications due to its biocompatibility, biodegradability and physical properties [12,13,14]. However, its hydrophobic nature limits its coupling with CuONPs made in aqueous solution. Herein, we propose to solve such limitations is by merging PCL with CuONPs synthesized in a non-aqueous reducing solvent to prepare a composite with degradability, and controllable mechanical properties proportioned by the PCL and with antimicrobial activity against pathogenic Candida species depending on the initial concentration of CuONPs dispersed alongside the fibers.

## Materials and methods

In this work, all the chemicals used were of analytical grade. The reagents copper nitrate (Cu(NO$_3$)$_2$), poly ε-caprolactone (average molecular weight of 80 k), gallic acid (≥97.5%), dimethylformamide (DMF) (≥98.5) and tetrahydrofuran (THF) (≥99.5%) were acquired from Sigma Aldrich. Three Candida species (*Candida albicans* ATCC MYA-2876, *Candida glabrata* ATCC 2001 and *Candida tropicalis* ATCC 750) used in this study were purchased from the American Type Culture Collection (ATCC), USA.

## Copper oxide nanoparticles preparation

Synthesis of CuONPs was carried out by a non-aqueous chemical method using copper nitrate as precursor and gallic acid as reducing agent according to [15]. Initial concentrations of copper nitrate (0 mM, 25 mM, 50 mM, 100 mM and 200 mM) were added into a solution of 7:3 of DMF and THF respectively; and then it was placed under magnetic stirring were gallic acid (0.02 M) in DMF:THF solution was added dropwise. The colour of the solution changed from blue to dark green. UV–Vis absorption spectra were measured with a variable wavelength between 100–900 nm at room temperature in a Cary100 spectrophotometer (Varian Corp.). Dynamic light scattering (DLS) was employed to measure particle size and distribution in an HORIBA SZ-100 Nanoparticle Analyzer. Raman analysis was carried out by a WITec's Raman spectrophotometer alpha300 R, with a 532 nm laser. Finally, a scanning electron microscopy (SEM, JEOLJSM-6400) operated at 20 kV was used to observe the morphologies of the PCL-CuONPs fibers.

## Preparation and fabrication of fiber composites

The next step was the preparation of fibers by the technique of electrospinning; the simple method consisted in forming a composite solution of the newly prepared CuONPs solutions with PCL 10% m/v under magnetic stirring at room temperature. The resulting viscous solution of PCL-CuONPs was deposited into a syringe connected to a 1.25 mm inner diameter steel needle connected to a high-voltage generator. A dense web of fibers was collected on an aluminum foil which served as the counter electrode. An electrical potential of 15 kV was used, the distance between the needle and the counter electrode was 10 cm, the feeding rate of the solution in syringe pump was 15–20 μL/min and the temperature in which the electrospinning assay was performed was 25 ºC.

## Preparation of fungi suspension

Candida species, *C. albicans*, *C. glabrata*, and *C. tropicalis* were used in this study. Isolates were cultured on Sabouraud Dextrose Agar (SDA) for 24 h at 37˚C before testing. The Antibiotic Susceptibility Testing by broth microdilution method was used to test all the species of Candida for the in vitro susceptibility to plain PCL and PCL-CuONPs composites according to the European Committee on Antimicrobial Susceptibility Testing (EUCAST). The inoculum was prepared by suspending five distinct colonies, ≥1 mm diameter in at least 3 mL of sterile distilled water. The suspension was homogenized for 15 seconds with a gyratory vortex mixer at approximately 2,000 rpm. Cell density was adjusted to the density of a 0.5 McFarland standard by measuring absorbance in a spectrophotometer (Varian Corp.) at a wavelength of 530 nm and adding sterile distilled water as required. Working suspension was prepared by a 1 in 10 dilution of the standardized suspension in sterile distilled water to yield $1–5 \times 10^5$ CFU/mL.

### *In vitro* antifungal susceptibility testing

The microdilution plates were inoculated with 200uL of the yeast working suspension into each well with the defined concentrations for PCL and PCL-CuONPs (25, 50, 100 and 200 mM). To inoculate the growth control wells 100 μL of sterile medium, with 100 μL of the same inoculum suspension were used. The antimycotic test for all Candida species and fibers was made in quadruplicate, and all data are expressed as mean values ± SE and analyzed by IBM SPSS Statistics 25. ANOVA and Tukey's multiple comparison test were used to carry out the statistic analyses, considering a p value of ≤0.05 as statistically significant. Finally, the

morphologies of the yeast cells exposed to PCL-CuONps fibers were observed with a scanning electron microscopy (SEM, JEOLJSM-6400) operated at 20 kV.

## Results and discussion

The formation of CuONPs was registered from the color change in the solutions, from blue to dark green, corresponding to the formation of CuONPs. UV-Vis analysis showed the plasmon band absorption in Fig 1A corresponding for CuONPs, according to the previous work of Varughese, G. et al [16]. which attributed the formation of cupric oxide nanoparticles to an excitonic absorption peak at 280 nm. Our analysis shows the absorption peaks increasing accordingly to the precursor salt concentration from 270 to 285 nm. The detected changes in

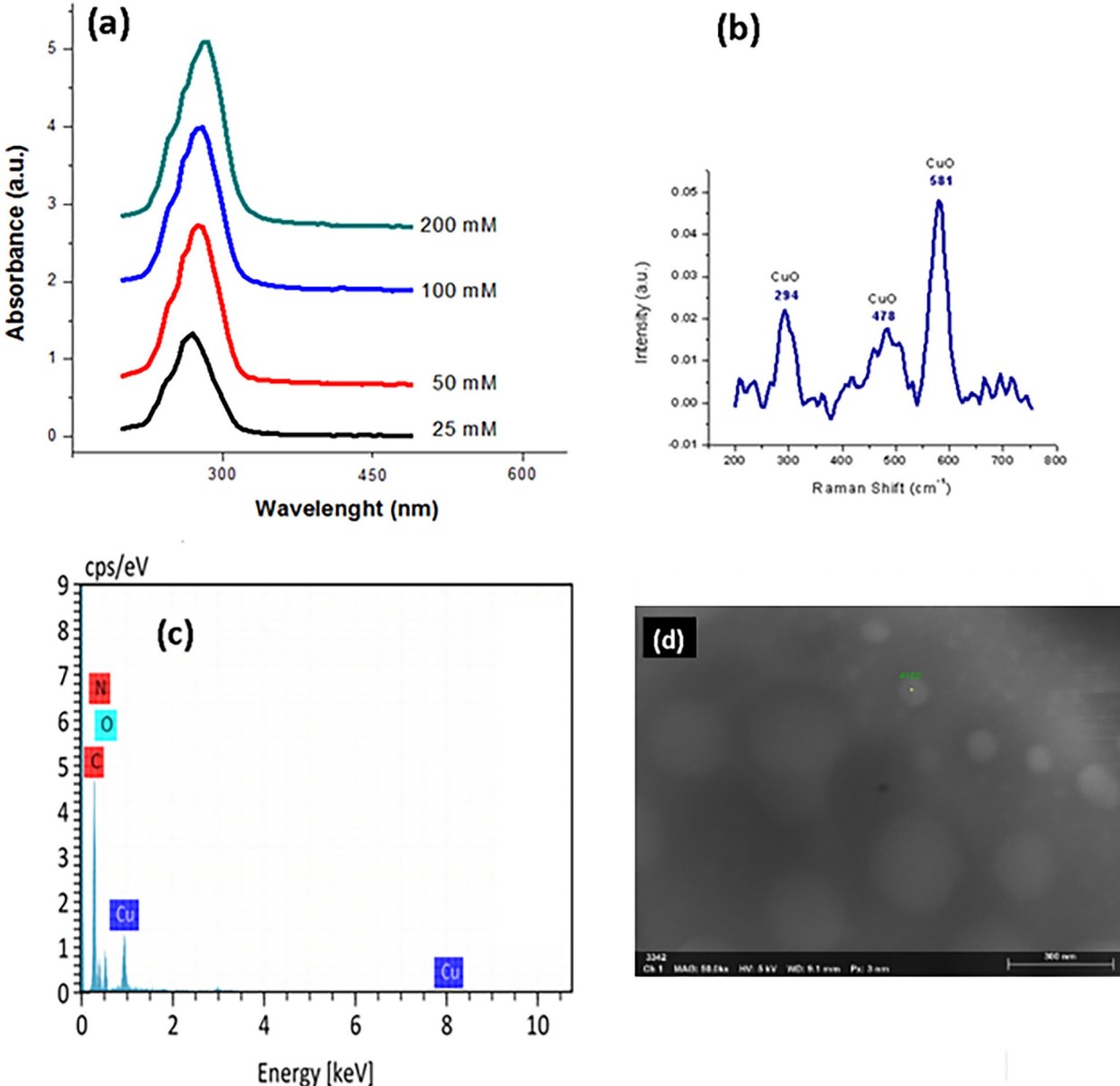

**Fig 1.** a) UV-Vis spectra, b) Raman spectra and c) EDX and d) SEM characterization of CuONPs.

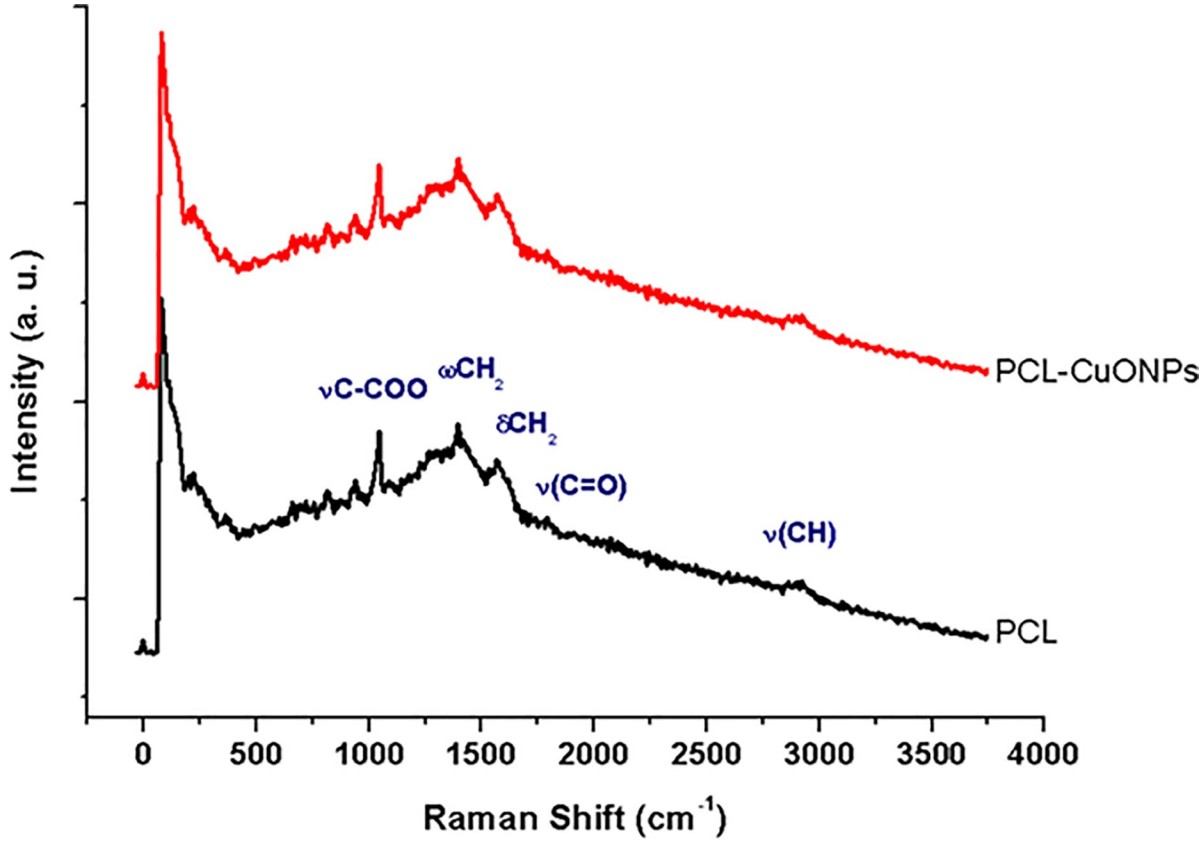

**Fig 2. Raman spectra of PCL and PCL-CuONPs fibers.**

the spectrum reflect the distinctive pattern of CuONPs formation by reducing copper ions with gallic acid present in reductive solution of DMF:THF. Further evidence for the synthesis of CuONPs is provided by the Raman spectra of the synthesized nanoparticles illustrated in Fig 1B. In Raman spectra are three bans in the sample, at 294, 581 and a very broad band from 400 to 600 cm$^{-1}$ which are characteristics bands for CuO. It was possible to discard the $Cu_2O$ presence in the films, due to spectra for $Cu_2O$ shows very different features at distinct bands at 150, 220 and 625 cm$^{-1}$. The non-appearance of peaks related to another single-phase of $Cu_xO_y$ was confirmed by the absence of $Cu_2O_3$, $Cu_2O$ and Cu planes in XRD, in agreement with Raman spectra only XRD patterns of the CuO are observed (according with JCPDS file No. 45–0937). The obtained nanoparticles observed from the micrograph majority were mostly spherical and some of them were agglomerated in SEM image of Fig 1C. Without any protecting agents, the general expectation would be that the nanoparticles would tend to agglomerate even more and that the particle sizes would be larger and more variable. However, there was noted only some variation in nanoparticle size. Most sizes of the particles ranged from 20 to 45 nm, and the average size was estimated at 35 nm for all concentrations according previous work [15]. Nanoparticle sizes were not directly proportional to the precursor salt concentration. The hydrodynamic diameters of CuONPs by DLS were measured showing uniform and well distributed sizes of CuONPs in the range of 88 ±97 nm. Moreover, the reaction was assumed to be complete due to the original quantities of precursor salts and reducing agents left no residue. EDX spectrum of CuONPs in Fig 1C shows carbon, oxygen, nitrogen and copper are the principal elements forming the sample.

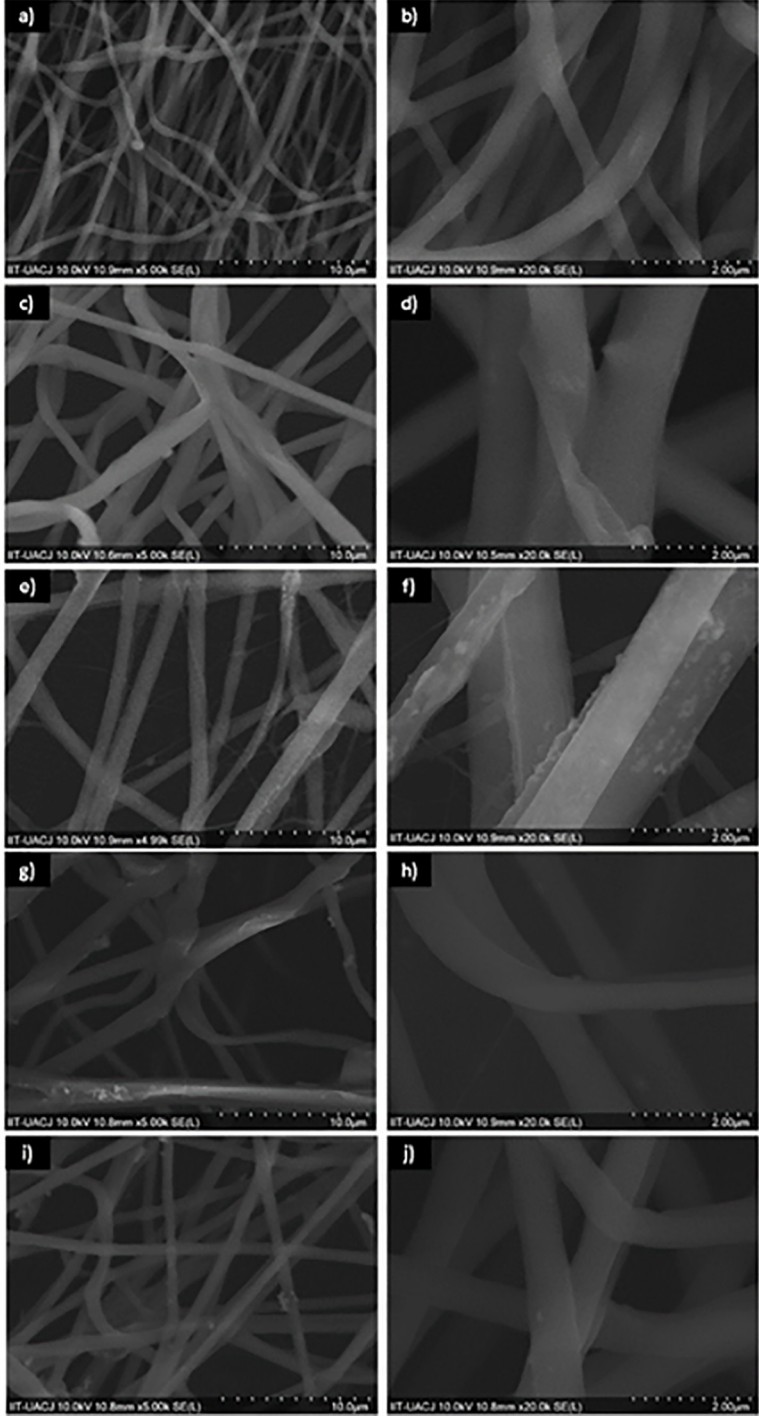

**Fig 3.** SEM images of PCL at 0 mM (a) 5,000 X and b) 20,000 X); PCL-CuONPs at 25 mM (c) 5,000 X and d) 20,000 X); PCL-CuONPs at 50 mM (e) 5,000 X and f)20,000 X); PCL-CuONPs at 100 mM (g) 5,000 X and h) 20,000 X) and PCL-CuONPs at 200 mM (i) 5,000 X and j)20,000 X) of coper nitrate used as precursor.

The Raman spectroscopy is a versatile method that allows studying the degree of crystalline and amorphous phases of composite membranes. The Raman spectrum of PCL and PCL-CuONPs is shown in the Fig 2. Bands assigned to the copper–oxygen mode (Cu–O) or to the ionic species are generally eclipsed by the polymer. Several narrow peaks at 934 cm$^{-1}$ ($vC$–

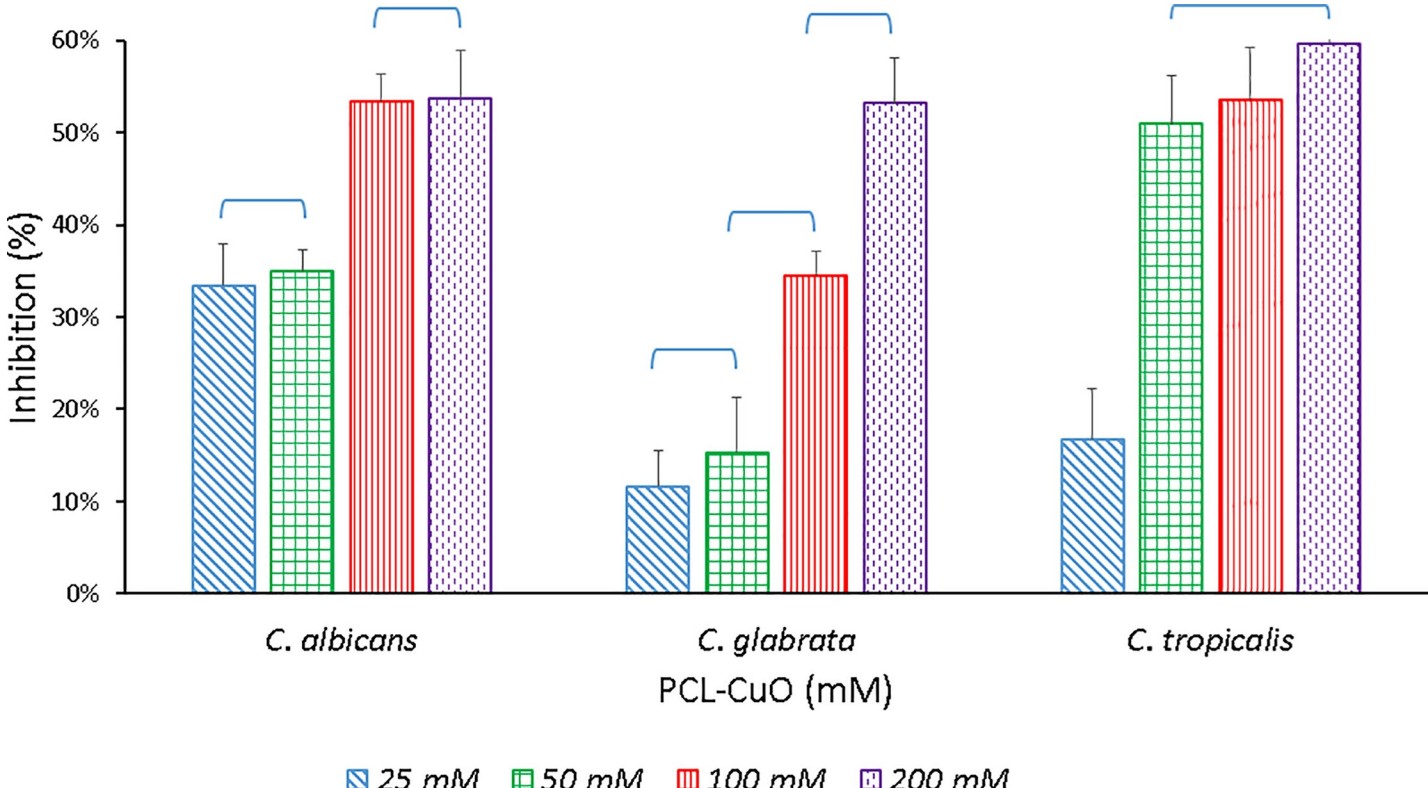

**Fig 4. Inhibition effect of PCL-CuONPs at 25, 50, 100 and 200 mM of Copper nitrate, used as precursor on Candida spp.**

COO), and others within the spectral ranges 1000–1150 cm$^{-1}$ (skeletal stretching), 1280–1350 cm$^{-1}$ (wCH$_2$), 1400–1500 cm$^{-1}$ (dCH$_2$) and 2800–3000 cm$^{-1}$ (vCH) are referred to the crystalline fraction of the PCL. The broad peak at 825 cm$^{-1}$ indicates that the amorphous phase is present in the PCL fibers [17].

SEM images in Fig 3 obtained from the PCL and PCL-CuONPs fibers scaffolds, which demonstrate a smooth surface, with no beads, precipitates or fractures. There is an evident relation between the concentration of CuONPs and the widening of the fibers, as the SEM results showed. The presence of this metal nanoparticles incremented the electrical charge, conductivity and viscosity on the solutions, resulting on an increase of the fibers diameter. The average diameters of PCL and PCL-CuONPs nanofibers were from 522±156 nm, 925±279 nm, 908±18 nm, 945±388 nm and 1082±329 nm for CuONPs contents of 0, 1, 10, 50 and 100 nM, respectively according previous data reported [15].

The antifungal capacity of the PCL-CuONPs fibers was evaluated through the EUCAST protocol with different species of the Candida genus. It was determined that PCL fibers does not exert inhibitory effects on Candida cells, as there was no difference between the growth of cultures treated with PCL compared to the control cultures. As seen in Fig 4 there is clearly an antifungal activity by the CuO nanoparticles from an initial concentration of 25 mM in all Candida species. The species of *C. albicans* and *C. glabrata* showed an inhibition in their growth of 53% in the maximum concentration of CuONPs used, while *C. tropicalis* had an even greater effect of up to 59%.

Fig 5 shows SEM images of *C. albicans* showing a clear difference in cell morphology when it is exposed only to the polymer fibers without the metal oxide, observing a defined and

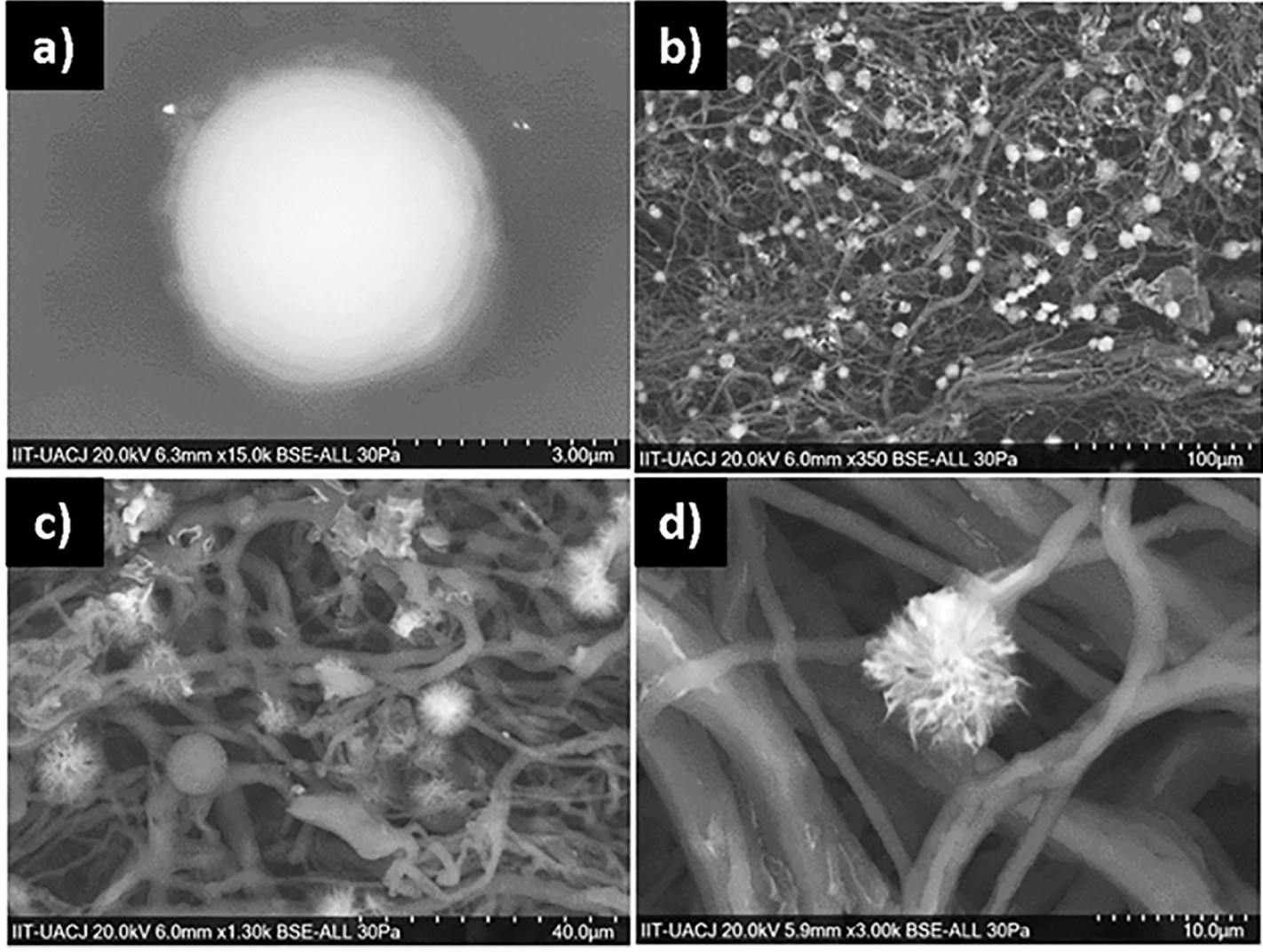

**Fig 5. Inhibition effect of PCL-CuONPs against C. albicans.** a) and b) are images for C. albicans exposed to plain PCL, and c) and d) are images of C. albicans exposed to PCL-CuONPs at 200 mM.

integral cell contour. In the rest of the images it is possible to distinguish that although some cells with their intact cell wall still adhere to the 200 mM PCL-CuONPs fibers, most show a severe damage to their structure, being observed deformed, distorted and contracted.

The SEM images of *C. glabrata* show the integrated cell wall of the yeast in the control conditions (Fig 6), and the damage caused by the maximum concentration of CuONPs, where damage such as the formation of pits or the rupture of the cell wall in the observed cells, are an indicative of an ongoing process of apoptotic cell death. *Candida albicans* and *C. glabrata*, are commensals of humans, and can be found in the oral cavity and the gastrointestinal tract of most healthy humans. The morphological flexibility of *C. albicans*, being a diploid, polymorphic fungus, presents a morphological flexibility that seems to play a role in several aspects of its pathogenicity. In contrast, the infection of *C. glabrata* seems to be independent of morphology due to it being strictly haploid [18].

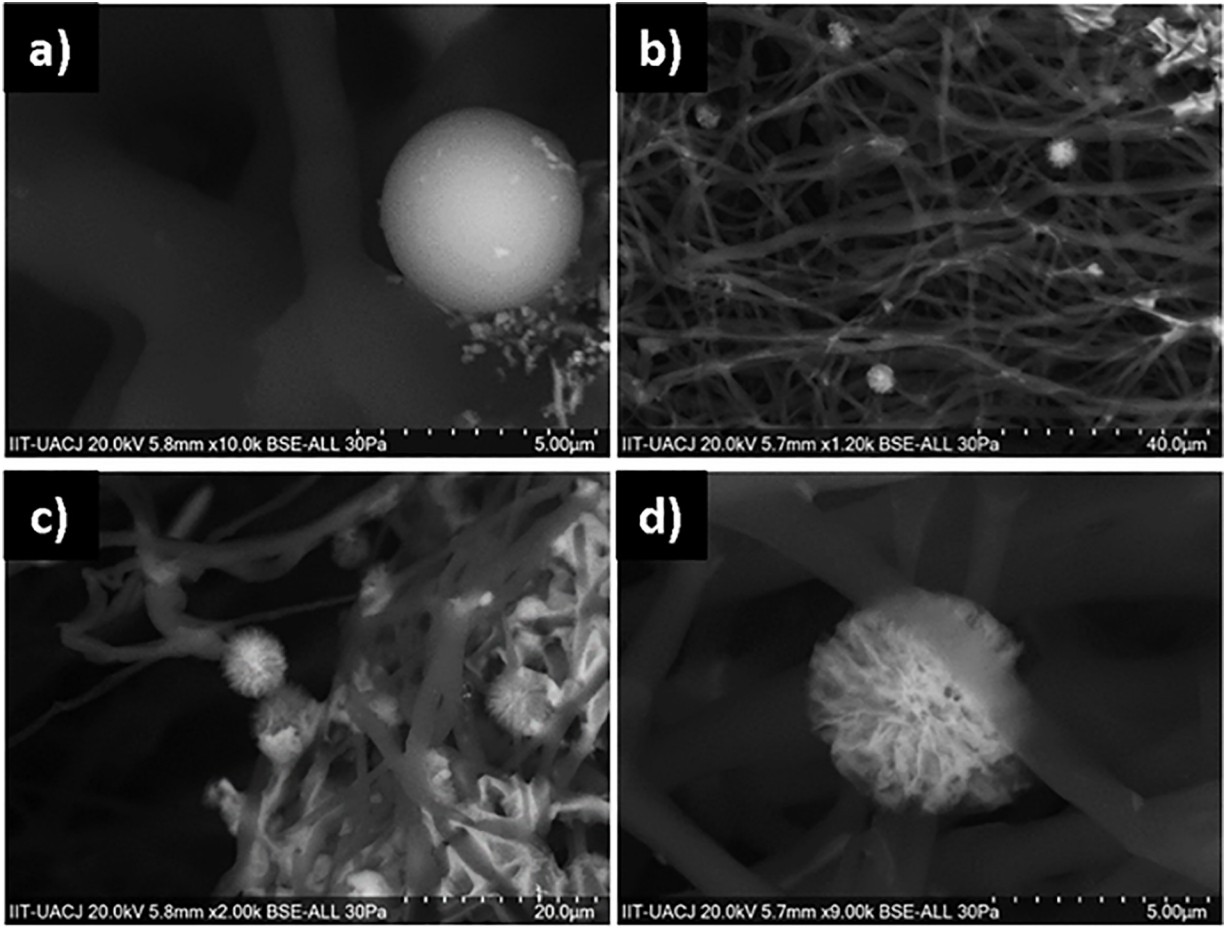

**Fig 6. Inhibition effect of PCL-CuONPs against *C. glabrata*.** a) and b) are images for *C. glabrata* exposed to plain PCL, and c) and d) are images of *C. glabrata* exposed to PCL-CuONPs at 200 mM.

From the concentration of 50 mM PCL-CuONPs, a clear difference in the inhibition in *C. tropicalis* is discernible, maintaining this percentage of inhibition along the subsequent concentrations of the nanoparticles. This species of Candida was the one that presented a greater inhibition with respect to the other species analyzed, which can be visualized in the images of SEM where we can observe first the spherical morphology of *C. tropicalis* in the control fibers, and the loss thereof, being found only debris of the organic material of the yeasts along the fibers (Fig 7).

Although fluctuations in the external environment can affect the unicellular species of Candida, these cells respond differently to adverse environmental conditions to carry out their growth. At present, there are serious efforts to discover compounds with promising antimicrobial activities. In this study, the extent of antifungal activity of PCL-CuONPs was studied further in three Candida species present in the human oral cavity. The effectiveness in the growth profiles of the organisms was determined based on the monitoring of changes in the optical density of fungal growth over time [5].

Previous studies testing conventional fungicidal agents have shown that *Candida albicans*, *Candida tropicalis*, and *C. glabrata* were more susceptible than other Candida species [19]. Antifungal activity depends on the cellular structure, cellular physiology, metabolism or degree of contact. Although antimicrobial mechanisms of nanoparticles are not comprehended

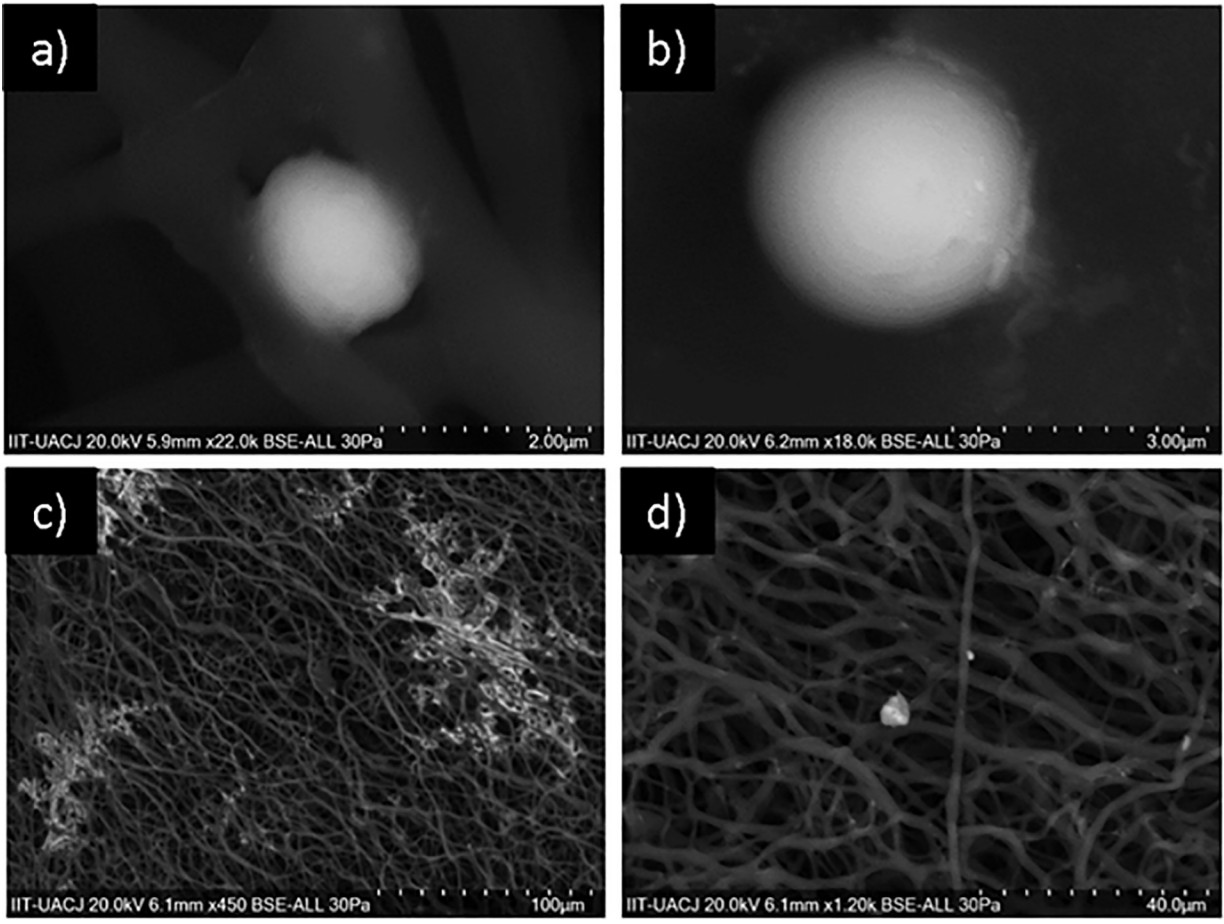

**Fig 7. Inhibition effect of PCL-CuONPs against C. tropicalis.** a) and b) are images of *C. tropicalis* exposed to plain PCL, and c) and d) are images *C. tropicalis* exposed to PCL-CuONPs at 200 mM.

throughout, it is proposed that they induce the creation of reactive oxygen species (ROS), cell membrane disruption, and mitochondrial and DNA damage. CuONPs generate potent cyto-toxicity by damaging DNA and epithelial cells of the respiratory tract due to oxidative stress [20]. The cytoxicity of CuONPs can be attributed to solubility by leaching, relatively high in biological media, with contact and interaction with biological fluids being likely. CuONPs release sufficiently toxic amounts of soluble copper in the biological environment. CuONPs penetrate the cell membrane followed by intracellular dissolution in active redox copper ions. The release of copper ions generates toxicity highly favored by nanometric size. The development of oxidative stress as a cause of the fungicidal activity agrees with the result of the oxidant test, where the oxidative capacity of CuONPs is observed [21].

Superficial fungal infections caused by Candida albicans require the dimorphic transition from yeast to mycelial form. In our study, the mycelia were significantly inhibited from extending and forming in the presence of CuONPs [22]. It is important to mention that in a biological environment the factors of pH, ionic strength, dissolved organic matter and bio-availability of molecular ligands would play an important role in the toxicity due to the leach-ing of CuONPs. Considering that Candida species are part of the normal flora in the human oral cavity, it is understandable that the yeast cells grow at a minimum speed, or at least the excessive growth of the candida species on the other microflora is decreased [5].

## Conclusions

PCl and PCL-CuONPs fibers with diameters ranging from 925 to 1080 nm were successfully obtained by electrospinning technique. Orientation, morphology and diameter were influenced by the increment on CuONPs concentration, with the smaller diameter present in samples prepared from low concentrated solutions. Candida growth was not affected by PCL fibers only; nevertheless, CuONPs presence and concentration determined the antifungal activity on the different species of Candida, even on the lowest concentration. Furthermore, their cellular morphology was affected by being exposed to PCL-CuONPs composites determining a dose-dependent activity. These composite fibers might enhance actual antimycotic therapies against multidrug-resistance fungi that represent a serious oral health issue.

## Acknowledgments

Thanks to PRODEP, Universidad Autónoma de Ciudad Juárez and CONACYT for supporting this investigation.

## Author Contributions

**Conceptualization:** Simón Yobanny Reyes-López.

**Investigation:** Antonio Muñoz-Escobar, Simón Yobanny Reyes-López.

**Project administration:** Simón Yobanny Reyes-López.

**Resources:** Simón Yobanny Reyes-López.

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
