## [Decision Letter · Decision Letter 0]

2 Sep 2019

PONE-D-19-18393

Antifungal susceptibility of oral Candida species to copper oxide nanoparticles on polycaprolactone fibers (PCL-CuONPs)

PLOS ONE

Dear Dr Reyes-López,

Thank you for submitting your manuscript to PLOS ONE. After careful consideration, we feel that it has merit but does not fully meet PLOS ONE’s publication criteria as it currently stands. Therefore, we invite you to submit a revised version of the manuscript that addresses the points raised during the review process.

We would appreciate receiving your revised manuscript by Oct 17 2019 11:59PM. To enhance the reproducibility of your results, we recommend that if applicable you deposit your laboratory protocols in protocols.io, where a protocol can be assigned its own identifier (DOI) such that it can be cited independently in the future. For instructions see: http://journals.plos.org/plosone/s/submission-guidelines#loc-laboratory-protocols

We look forward to receiving your revised manuscript.

Kind regards,

Amitava Mukherjee, ME, Ph.D.

Academic Editor

PLOS ONE

Journal Requirements:

Additional Editor Comments (if provided):

Reviewers' comments:

Reviewer's Responses to Questions

**Comments to the Author**

1. Is the manuscript technically sound, and do the data support the conclusions?

Reviewer #1: Partly

2. Has the statistical analysis been performed appropriately and rigorously? 

Reviewer #1: N/A

3. Have the authors made all data underlying the findings in their manuscript fully available?

Reviewer #1: No

4. Is the manuscript presented in an intelligible fashion and written in standard English?

Reviewer #1: No

5. Review Comments to the Author

Reviewer #1: In general, the manuscript was written very poorly. Likewise, explanations of the results are rather shallow or not explained at all.

1. Some phrases in the conclusion is not supported in the results or vague

(a) “…diameter were affected by the increment of CuONPs concentration.” This is not supported by the data You only show with and without CuONP, no “increment” or varying CuONP concentrations was presented.

(b) “….their cellular morphology was demonstrated to be affected by PCL-CuONPs composites determining a dose-dependent activity”. Please rephrase, the sentence construction is vague.

2. I have not seen the raw data of the UV-VIS spectra . (No attached supplementary data.)

3. The use of English language needs improvement. Many sentences are vague. You need the help of a professional grammar editor.

4. The SEM images do not support that the fibers are nanometer in thickness. Although there is no “nanofiber” word in the title, the word “nanofiber” was used to describe the fiber samples in some parts of the manuscript (e.g., abstract, results and discussion, conclusion).

5. Title: You can remove the word “oral” in the title, “Candida Species” is enough.

6. Abstract: Line 13: “Polycaprolactone-silver…”. Silver was written instead of Copper.

7. Abstract: Line 16-17: “the second step involved the simple addition of polycaprolactone before electrospinning process”. The sentence is vague. Please rephrase. I think you mean that the second step involved electrospinning using polycaprolactone maybe as matrix of the CuO.

8. Lines 82-86 (Methodology: Copper oxide nanoparticle preparation): Some phrases sentences are vague and confusing. Simplify the sentence construction.

9. The caption of Figure 1 is poorly written. The figure labels (a,b,c) should be written also in the “caption of the figure” to concretely match its label/description.

10. Figure 1a is not an UV-VIS spectrum but rather the intensity of a certain peak from the UV-VIS spectrum. Rewrite the description more concretely.

11. Figure 1a: Is the x-axis really CuONP concentration? Or it is the initial copper nitrate concentration used?

12. Figure 1a: Does the label “270”, “275”, “280”, “285” pertains to the peak absorbance or wavelength position of the peak? The unit of your y-axis is “absorbance” while in the manuscript the unit is nm.

13. The explanation of Figure 1a in the manuscript is very shallow. What possible cause the shift? Instead of just saying “The observe changes in the spectrum reflects the characteristic pattern of CuONP formation…”.

14. Figure 1b is not merely the “size” but rather the frequency distribution of the samples diameter as measured by DLS.

15. The concentration labels in Figure 1b are not specified that is the initial concentration of copper nitrate.

16. “Figure 1a” and “Figure 1b” are not “called/mentioned” in the body of the manuscript

17. What is the purpose of having Raman Spectrum in your paper? You have not detected presence of CuO using Raman Spectroscopy. Likewise, you have already used commercially available PCL. Do you need to still identify that you have PCL using Raman Spectroscopy? Or do you need to identify the crystalline of your sample? Is determining the crystallinity relevant to your study?

18. Line 159-162: PCL is semicrystalline polyester. However, are you sure that what you have mentioned to be “narrow” peaks really are narrow to be able to signify “crystallinity” of PCL? Do you need to specify the crystalline of your sample in your paper?

19. Line 162. “The broad peak at 825 cm-1 indicates that the amorphous phase is present in the PCL fibers (Baranowska).” Where is the broad peak at 825 cm-1 (Figure 1c)? Do you forget to label or it is not present ?

20. Based from Figure 2b, the average diameter of the synthesized copper particles decreases with the increase in initial copper nitrate concentration. Briefly explain the possible reason of this trend.

21. Your Figure 2 has labels “a” and “b” but the figure caption does not have corresponding labels.

22. Line 165: “….free area of beads, precipitates and fractures.” Rephrase this phrase, I think you mean “with no beads, precipitates and fractures”.

23. Line 167-169: “Results obtained by electron microscopy showed the existing relation between size distribution and concentration of CuO nanoparticles.” This statement has no supporting explanations.

24. Figure 3: “PCL-CuO (mM) 25mM, 50mM….” The “mM” pertains to the initial copper nitrate concentration used in fabricating CuO, and not the concentration of CuO in PCL fiber? If so, the label “PCL-CuO (mM) might be misleading”.

25. What do you mean by the “inverted bracket” in Figure 3? Please specify in its caption.

26. The captions of Figures 4,5 and 6 are vague. They do not give specifications of what is a, b, c, and d”. Please rewrite the captions.

27. Figures 4,5, and 6 were described in the manuscript without identifier of which image (a,b,c or d) they are taking about. For example, they should have state that Figure 4a (instead of just Figure 4) is the SEM image of C.Albican which was exposed to polymer fiber without the CuO.

28. Line 194: “The SEM images of C. glabrata, as well as C. albicans, show the integrated cell wall of the yeast in the control conditions in Fig. 5 ” No need to write “as well as C. albican” since it is not part of Figure 5. If you want to write “as well as C. Albicans”, rephrase the sentence.

6. PLOS authors have the option to publish the peer review history of their article (what does this mean?). If published, this will include your full peer review and any attached files.

Reviewer #1: No

---

## [Author Response · Author response to Decision Letter 0]

19 Oct 2019

All corrections were made, information and images requested were added.

The corrections were made in the manuscript, figures and discussion were attached, the summary as well as the conclusions were modified

---

## [Decision Letter · Decision Letter 1]

12 Nov 2019

PONE-D-19-18393R1

Antifungal susceptibility of Candida species to copper oxide nanoparticles on polycaprolactone fibers (PCL-CuONPs)

PLOS ONE

Dear Dr Reyes-López,

Thank you for submitting your manuscript to PLOS ONE. After careful consideration, we feel that it has merit but does not fully meet PLOS ONE’s publication criteria as it currently stands. Therefore, we invite you to submit a revised version of the manuscript that addresses the points raised during the review process. 

We would appreciate receiving your revised manuscript by Dec 27 2019 11:59PM. To enhance the reproducibility of your results, we recommend that if applicable you deposit your laboratory protocols in protocols.io, where a protocol can be assigned its own identifier (DOI) such that it can be cited independently in the future. For instructions see: http://journals.plos.org/plosone/s/submission-guidelines#loc-laboratory-protocols

We look forward to receiving your revised manuscript.

Kind regards,

Amitava Mukherjee, ME, Ph.D.

Academic Editor

PLOS ONE

Reviewers' comments:

Reviewer's Responses to Questions

**Comments to the Author**

1. If the authors have adequately addressed your comments raised in a previous round of review and you feel that this manuscript is now acceptable for publication, you may indicate that here to bypass the “Comments to the Author” section, enter your conflict of interest statement in the “Confidential to Editor” section, and submit your "Accept" recommendation.

Reviewer #1: (No Response)

2. Is the manuscript technically sound, and do the data support the conclusions?

Reviewer #1: Yes

3. Has the statistical analysis been performed appropriately and rigorously? 

Reviewer #1: N/A

4. Have the authors made all data underlying the findings in their manuscript fully available?

Reviewer #1: No

5. Is the manuscript presented in an intelligible fashion and written in standard English?

Reviewer #1: No

6. Review Comments to the Author

Reviewer #1: General Review:

The manuscript improved.

1. Line 165: “only XRD patterns of the CuO are observed”… the actual XRD pattern was not available.

Minor concerns:

Abstract

1. Line 23: there is a word “STEM images”, however, you have not discussed about the STEM images in the result and discussion. You only discuss about the SEM images.

2. Line 14: You can remove the word “capacity”.

3. Line 16: the word “polymer” is redundant since polycaprolactone is already a polymer. In the body of your work, you have described your sample as polycaprolactone fiber. Thus, “fiber” is better than “film”.

4. Line 17: “Polycaprolactone-silver fibers (PCCl-CuONPS) were prepared by reduction in-situ method of Cu+2 ions by gallic acid in ….”. Change the word “silver” to “copper”. The sentence is confusing. You are trying to incorporate two different concepts in one paragraph. Your method can be divided into two major steps: (a) Synthesis of CuO particles, then (b) incorporation of CuO particles on polycaprolactone fiber.

5. Line 20-21: Please rephrase. I think it is either “Raman spectra shows that you have synthesized copper oxide (CuO)” or “Raman spectra provide information about the nature of your synthesized sample”.

6. Line 25: “Dynamic light scattering…. showed uniform CuONps in a range of 88�11nm”. Is “88�11nm” a range? In the result and discussion, your range is 88-97nm. “P” is capital letter in the word “CuONps”.

Materials and Method

1. Line 90: Place comma between “Poly �-caprolactone (average molecular weight of 80K)” and “Gallic acid”.

Results and Discussion

1. Line 63: Rephrase the word “the not formation…”

2. Line 164:” What is “DRX”? Is this “XRD”?

3. Line 164: What is “whit”? Is this “with”?

4. Line 214-216: Indicate what is 0 mM, 25mM…? It is mM of Copper nitrate, used as precursor?

5. Captions of Figures 5,6 and 7: Indicate what is 200 mM? It is mM of Copper nitrate, used as precursor?

6. Figures 5 and 6” Change the word “Amplified images”? Based on the magnification, it is not amplified image but demagnified imaged.

7. PLOS authors have the option to publish the peer review history of their article (what does this mean?). If published, this will include your full peer review and any attached files.

Reviewer #1: No

---

## [Author Response · Author response to Decision Letter 1]

16 Jan 2020

The corrections were made in the manuscript, paragraphs, figures and discussion were attached and modified, the summary as well as the conclusions were modified

All data is in the manuscript and can be shared

---

## [Editor Report · Decision Letter 2]

27 Jan 2020

Antifungal susceptibility of Candida species to copper oxide nanoparticles on polycaprolactone fibers (PCL-CuONPs)

PONE-D-19-18393R2

Dear Dr. Reyes-López,

We are pleased to inform you that your manuscript has been judged scientifically suitable for publication and will be formally accepted for publication once it complies with all outstanding technical requirements.

With kind regards,

Amitava Mukherjee, ME, Ph.D.

Academic Editor

PLOS ONE
---

## [Editor Report · Acceptance letter]

10 Feb 2020

PONE-D-19-18393R2 

Antifungal susceptibility of *Candida* species to copper oxide nanoparticles on polycaprolactone fibers (PCL-CuONPs) 

Dear Dr. Reyes-López:

I am pleased to inform you that your manuscript has been deemed suitable for publication in PLOS ONE. Congratulations! Your manuscript is now with our production department. 

With kind regards,

on behalf of

Professor Dr. Amitava Mukherjee 

Academic Editor

PLOS ONE